# Beneficial Effects of *In Vitro* Reconstructed Human Gut Microbiota by Ginseng Extract Fermentation on Intestinal Cell Lines

**DOI:** 10.3390/microorganisms13010192

**Published:** 2025-01-17

**Authors:** Margherita Finazzi, Federica Bovio, Matilde Forcella, Marina Lasagni, Paola Fusi, Patrizia Di Gennaro

**Affiliations:** 1Department of Biotechnology and Biosciences, University of Milano-Bicocca, 20126 Milano, Italy; m.finazzi7@campus.unimib.it (M.F.); federica.bovio@unimib.it (F.B.); matilde.forcella@unimib.it (M.F.); paola.fusi@unimib.it (P.F.); 2Department of Earth and Environmental Sciences, University of Milano-Bicocca, 20126 Milano, Italy; marina.lasagni@unimib.it

**Keywords:** HGM, microbiota reconstruction, prebiotics, ginseng, probiotics, functional food, nutraceuticals, oxidative stress, antioxidant effect

## Abstract

Oxidative stress caused by reactive oxygen species (ROS) affects the aging process and increases the likelihood of several diseases. A new frontier in its prevention includes bioactive foods and natural extracts that can be introduced by the diet in combination with specific probiotics. Among the natural compounds that we can introduce by the diet, *Panax ginseng* extract is one of the most utilized since it contains a vast number of bioactive molecules such as phenolic acids, flavonoids, and polysaccharides that have been shown to possess antioxidant, anti-ageing, anti-cancer, and immunomodulatory activity. In this work, the ability of a *P. ginseng* extract in combination with a probiotic formulation was taken into consideration to evaluate its effects on the modulation of *in vitro* reconstructed human gut microbiota (HGM). After evaluating the growth of the individual strains on the ginseng extract, we tested the *in vitro* reconstructed HGM setup (probiotics, minimal core, and whole community) using 2% *w*/*v* ginseng as the only carbon and energy source. The probiotic strains reached the highest growth, while the minimal core and the whole community showed almost the same growth. Specifically, the presence of the ginseng extract favors *L. plantarum* and *B. animalis* subsp. *lactis* among the probiotics, while *B. cellulosilyticus* prevails over the other strains in the minimal core condition. In the presence of both probiotics and minimal core strains, *L. plantarum*, *B. animalis* subsp. *lactis*, and *B. cellulosilyticus* reach the highest growth values. The bacterial metabolites produced during ginseng extract fermentation in the three conditions were administered to human intestinal epithelial cells (HT-29) to investigate a potential antioxidant effect. Remarkably, our results highlighted a significant reduction in the total ROS and a slightly reduction in the cytosolic superoxide anion content in HT-29 cells treated with bacterial metabolites deriving from ginseng extract fermentation by the whole community.

## 1. Introduction

The human intestine is inhabited by high microbial biodiversity constituted by hundreds of bacterial species [1]. Analysis of 16S rRNA gene sequences highlighted that a healthy human colon is dominated by two bacterial phyla, the Bacillota (previously Firmicutes) and Bacteroidota (previously Bacteroidetes), representing up to 90% of the total gut microbiome [2,3]. The remaining 10% of the human gut microbiota (HGM) is composed of Pseudomonadota (Proteobacteria), Actinomycetota (Actinobacteria), and Verrucomicrobiota (Verrucomicrobia) [1]. This complex bacterial community plays a critical role in several host physiological processes, such as playing a fundamental role in digestion and metabolism, protection from pathogens, modulation of the immune system, controlling epithelial cell proliferation and antioxidant responses, and influencing brain–gut communication [4]. All of the primary functions carried out by HGM result from the complexity of the interactions established between the different bacterial populations. Indeed, microorganisms can communicate with the host and each other by producing diverse metabolites and molecules [5]. Among these, short-chain and branched-chain fatty acids (SCFAs and BCFAs) stand out for their beneficial effects on the host [5,6,7].

Diet exerts a significant impact on the composition of the intestinal microbiota and consequently can affect human health [1]. Components of the diet, principally polysaccharides, such as pectin, inulin, fructooligosaccharides (FOS), or xylan, are not altered or absorbed in the gut and can be degraded by different bacterial populations through a cross-feeding mechanism [8,9]. For example, members of the phylum Bacteroidota can break down inaccessible plant-derived carbohydrates introduced by the diet, mucin-associated glycans, and host polysaccharides, making them more available to the intestinal community thanks to the highest range of polysaccharide utilization loci (PUL) [10,11]. Furthermore, the *Bifidobacterium* genus is capable of utilizing simpler oligosaccharides that are readily available and releasing sugar monomers that could be used by other bacteria [12].

Single strains have been used to study the ability of gut microbes to degrade complex polysaccharides [1]. However, the intricate interplay between human gut microbiota bacteria is fundamental in maintaining intestinal homeostasis and beneficial effects. Indeed, recently, an increasing number of studies have focused on the development of *in vitro* models of the HGM to assess the effects of various dietary components. The HGM models available range from deep-well plates to super-controlled bioreactors, which can operate in batch, semi-continuous, and continuous modes, with single or multi-stages [13]. Their dependability is deeply associated with the viability and the performance of the bacterial inoculum. In particular, synthetic defined communities are utilized to understand the cause-and-effect relationships between the administration of a carbon source and the modulation of the human gut microbiota [14]. Steimle et al. [15] employed a 14-member synthetic human gut microbiome (composed of *Bacteroides ovatus*, *Bacteroides uniformis*, *Bacteroides thetaiotaomicron*, *Bacteroides caccae*, *Barnesiella intestinihominis*, *Roseburia intestinalis*, *Eubacterium rectale*, *Faecalibacterium praustnizii*, *Marvinbryantia formatexigens*, *Clostridium symbiosum*, *Colinsella aerofaciens*, *Escherichia coli* SH, *Akkermantia muciniphila*, and *Desulfovibrio piger*) to study the effects of dietary concentrated raw fibers. Shetty et al. [16] designed a synthetic microbiome termed the Mucin and Diet-based Minimal Microbiome (MDb-MM) to identify human gut microbes’ dynamic metabolic interactions and trophic roles. De Giani et al. [17] reconstructed a *minimal core* model of the gut microbiota composed of four strains (*Bacteroides cellulosilyticus*, *Clostridium symbiosum*, *Flavonifractor plautii*, and *Escherichia coli* ATCC 25922) to assess the effects of a prebiotic Maitake extract and three probiotic strains (*Lactiplantibacillus plantarum*, *Lactobacillus acidophilus*, and *Bifidobacterium animalis* subsp. *lactis*).

Among the compounds that we can introduce by the diet, *Panax ginseng* is one of the most studied since it contains different bioactive components such as phenolic acids (gallic acid, caffeic acid, coumaric acid, salicylic acid, and cinnamic acid), flavonoids, polysaccharides, amino acids, phytosterol, ginseng oil, and specific vitamins [18]. Each part of the ginseng plant has a unique ginsenoside profile, meaning the various parts likely exert different pharmacological effects [19]. The roots of this plant are traditionally used as herbal medicines for the treatment of human diseases [19,20]. Indeed, root extracts possess anti-stress, anti-diabetic, anti-inflammatory, antioxidative, anti-ageing, anti-cancer, and immunomodulatory activity [20]. Recently, some studies have shown that the berry of *Panax ginseng* has a much stronger pharmacological activity than its root [19,21]. Ginseng polysaccharides have attracted much attention recently because of their abundance and diverse known bioactivities, such as antioxidant activity [22]. Reactive oxygen species (ROS) are chemically active compounds that can cause cellular damage. Usually, ROS are produced during cellular metabolism, mainly from the oxidative phosphorylation process that occurs in the mitochondrial electron transport chain. Oxidative stress accumulation can lead to metabolic disorders such as cancer, ageing, inflammation, and neurodegeneration [23]. The cells have enzymatic and non-enzymatic antioxidative defense systems to compensate for the ROS concentration and maintain equilibrium [21,24]. Plant-derived polysaccharides are shown to possess free radical scavenging activities [20].

The goal of the present work is to evaluate the effects of a ginseng extract as a prebiotic [21] and a probiotic formulation on the modulation of an *in vitro* reconstructed synthetic gut microbiota [17]. The activity of the metabolites from microbial ginseng extract fermentation was then evaluated on human intestinal epithelial cells to investigate a potential antioxidant effect.

## 2. Materials and Methods

### 2.1. Extraction and Characterisation of Ginseng Bioactive Molecules

The *Panax ginseng* berries extract (*P. ginseng* C.A Meyer, sold as “Panaxolyde”) was supplied by Flanat Research Italia Srl (Rho, Italy). The method for obtaining the extract from the berries of *Panax ginseng* shown in Figure 1 is reported in De Giani et al. [21].

The principal components of the ginseng powder were characterized. The protein content, polyphenol content, ginsenoside content, and polysaccharides content were determined with the methods described in De Giani et al. [21]. The components are reported in Table 1.

### 2.2. Bacterial Strains and Culture Conditions

The bacterial strains used in this study are listed in Table 2. The probiotics (*Lactiplantibacillus plantarum*, *Lactobacillus acidophilus*, and *Bifidobacterium animalis* subsp. *lactis*) were supplied by Roelmi HPC (Origgio, Italy) [25]. The minimal core microbiota strains *Bacteroides cellulosilyticus*, *Clostridium symbiosum*, and *Flavonifractor plautii* belong to BEI Resources (NIAID, NIH collection, as part of the Human Microbiome Project), and *Escherichia coli* ATCC 25922 derives from the American Type Culture Collection (ATCC, Manassas, VA, USA).

*Lactobacillus* and *Bifidobacterium* strains were maintained in De Man, Rogosa, and Sharp medium (MRS) (Conda Lab, Madrid, Spain) with the addition of 0.03% L-cysteine (Merk, Milano, Italy); minimal core microbiota strains were maintained in Reinforced Clostridia Medium (RCM) (Conda Lab, Madrid, Spain), with the addition of 0.03% L-cysteine and 0.01 g/L of hemin (Hemin chloride, Cayman Chemical, Ann Arbor, MI, USA). All the strains were grown at 37 °C in anaerobic conditions obtained by Anaerocult GasPack System (Merck, Darmstadt, Germany).

The experiments were performed in a modified MRS as mMRS supplemented with 2%, 1.08%, 0.18%, and 0.1% *w*/*v* of ginseng extract, pectin (pectin from apple, poly-D-galacturonic acid methyl ester, Merck, Milano, Italy), galactose (D-(+)-Galactose, Merck, Milano, Italy), and galacturonic acid (D-(+)-Galacturonic acid monohydrate, Merck, Milano, Italy), respectively [21]. The substrates were prepared as described by De Giani et al. [21]. The growth experiments with single strains were conducted in a final volume of 1 mL in 24 multiwells (24 wells, SPL Lifesciences, Pocheonsi, Korea). Each strain’s initial optical density at 600 nm (OD_600 nm_) was 0.1. All of the strains were grown statically in anaerobiosis (Anaerocult GasPack System, Merck, Darmstadt, Germany) at 37 °C for 48 h, and the OD_600 nm_ was measured.

### 2.3. Experiments in Batch Fermentation System

Pre-cultures of 72 h for the minimal core microbiota strains and of 48 h for the probiotics were prepared at 37 °C in anaerobic conditions before the setup of the experiments [17].

The three consortia setups (probiotics, minimal core, and whole community) were cultured in a 400 mL batch reactor (Colaver, Vimodrone, Italy), using mMRS (negative control) or mMRS supplemented with 2% ginseng extract as a carbon source. In all of the experimental conditions, the batch reactor was stirred at 60 rpm and maintained under anaerobic conditions by fluxing nitrogen 5.0 (Sapio, Monza, Italy) for 48 h at 37 °C. All of the bacterial strains were added with an initial OD_600 nm_ of 0.05 [17].

The different experimental setups were followed, as reported by De Giani et al. [17]. Briefly, 10 mL samples were taken every 8 h (T0, T8, T16, T24, T32, and T48). Nitrogen was flushed for 10 min before and after the sampling to maintain anaerobiotic conditions. Overall microbial growth was evaluated by OD_600_ nm. Culture broth was analyzed for microbial metabolites’ production. Bacterial cells were stored for DNA extraction and bacterial species quantification.

### 2.4. Total DNA Extraction

Total DNA was extracted from single bacterial cultures at a concentration of 10^8^ CFU/mL to construct a standard curve to title the strains by qPCR analyses [26,27]. Total DNA was obtained using the Ultraclean Microbial DNA Isolation Kit (Qiagen, Milano, Italy) following the protocol provided by the manufacturer.

The total DNA of the three fermentation setups was obtained using the Stool Nucleic Acid Isolation Kit (Norgen Biotek Corp., Thorold, ON, Canada) following the protocol provided by the manufacturer with some modifications [27].

The concentrations and purity of the DNA were analyzed spectrophotometrically (NanoDrop One Microvolume UV–Vis Spectrophotometer, ThermoFisher Scientific, Monza, Italy).

### 2.5. Modulation of the Strain Abundances of the In Vitro Reconstructed HGM Through qPCR

The single strain modulation was followed through qPCR reactions using PCR Real-Time StepOne Plus (Applied Biosystems, Monza, Italy) and the PowerUp SYBR Green Master Mix (Applied Biosystems, Monza, Italy) with species-specific primers. The bacterial counts/mL of each strain were calculated using the absolute quantification method, which utilizes a calibration curve. The average slope and y-intercept of each standard curve were determined by regression analyses and used to calculate the bacterial counts/mL for each bacterial target. The species-specific primer sets utilized in this study are reported by De Giani et al. [17] and in Appendix A [26,27,28,29,30,31]. Different qPCR programs were employed, as described by De Giani et al. [17]. Each DNA sample was analyzed in triplicate.

### 2.6. Extraction and Characterisation of Microbial Metabolites

After acidification at pH 2 (6 M HCl) of the cultured broth, bacterial metabolites were extracted as reported by De Giani et al. [21]. A 1:1 ratio of Ethyl Acetate (Merck, Milano, Italy) was employed for the extraction.

The metabolite analysis was carried out with a gas chromatograph (Technologies 6890 N Network GC System, Agilent Technologies, Santa Clara, CA, USA) equipped with a mass selective detector (5973 Network, Agilent Technologies, Santa Clara, CA, USA). The analyses were performed as described by De Giani et al. [21]. All of the samples were injected three times. The obtained chromatograms and mass spectra were interpreted by comparison with the National Institute of Standards and Technology (NIST) library and with injected standard molecules (Merck, Milano, Italy).

### 2.7. In Vitro Experiments on Human Cell Lines

#### 2.7.1. HT-29 Cell Line Maintenance and Viability Assay

The colon cancer cell line HT-29 (ATCC^®^ HTB-38™) was cultured in DMEM medium supplemented with heat-inactivated 10% FBS, 2 mM L-glutamine, 100 U/mL of penicillin, and 100 µg/mL of streptomycin and maintained at 37 °C in a humidified 5% CO_2_ incubator. The HT-29 cell line was verified by short tandem repeat profiles generated by the simultaneous amplification of multiple short tandem repeat loci and amelogenin (used for gender determination).

For the evaluation of cell viability in the presence of the extracts, cells were seeded in 96-well microtiter plates at a density of 1 × 10^4^ cells/well and 24 h later treated with 0.1 and 0.5 mg/mL for each extract. After 48 h, cell viability was assessed through an *in vitro* MTT-based toxicology assay kit (Merck KGaA, Darmstadt, Germany), by replacing the medium with a complete medium without phenol red, and adding 10 µL of 5 mg/mL MTT (3-(4,5-dimethylthiazol-2)-2,5-diphenyltetrazolium bromide) solution to each well. After 2 h of incubation, the resulting formazan crystals were solubilized with 10% Triton-X-100 in acidic isopropanol (0.1 N HCl) and absorbance was recorded at 570 nm with a micro plate reader. The results were expressed as the mean values ± ES of at least three independent experiments.

All of the reagents for cell culture were supplied by EuroClone (EuroClone S.p.A, Pero, Italy).

#### 2.7.2. ROS Detection

Intracellular reactive oxygen species (ROS) were detected via the oxidation of 2′,7′-dichlorofluorescin diacetate (H_2_DCFDA) and dihydroethidium (DHE) (Merck KGaA, Darmstadt, Germany). Cells were seeded in 96-well black microtiter plates at a density of 1 × 10^4^ cells/well and cultured in complete medium for 24 h.

For total cytosolic ROS, seeded cells were incubated with 5 µM H_2_DCFDA in PBS (10 mM K_2_HPO_4_, 150 mM NaCl, pH 7.2) for 30 min in the dark at 37 °C; then, they were rinsed in PBS and treated for 4 h with 0.1 and 0.5 mg/mL for each extract or with TBHP 100 µM, as a positive control. Fluorescence (λem = 485 nm/λex = 535 nm) was measured using a fluorescence microtiter plate reader (VICTOR X3, PerkinElmer, Akron, OH, USA), whereas for the cytosolic superoxide anion (O_2_•^−^) content, seeded cells were first incubated with 10 µM DHE in complete medium for 30 min in the dark at 37 °C and then 0.1 and 0.5 mg/mL for each extract were added to perform a 4 h treatment. Subsequently, cells were washed with PBS and fluorescence (λex = 480–520 nm/λem = 570–600 nm) was measured using a fluorescence microtiter plate reader (VICTOR X3, PerkinElmer, Akron, OH, USA). Antimycin A at a final concentration of 150 µM was used as a positive control.

All of the fluorescence measurements were normalized against the total protein amount, determined by a Bradford assay [32].

### 2.8. Statistical Analyses

The data derived from the bacteria growth are shown as the mean values ± standard error. The statistical significance was assessed through Student’s *t* test and was defined as * *p*-value < 0.05, ** *p*-value < 0.01, or *** *p*-value < 0.001. Bacterial counts/mL resulting from real-time q-PCR analysis are reported as the mean values ± standard error. The statistical significance was assessed through the Wilcoxon Signed Rank test and was defined as * *p*-value < 0.05, ** *p*-value < 0.01.

The data derived from oxidative stress analysis on the intestinal human cell line are represented as the mean ± standard error. Statistically significant results are highlighted with **** *p*-value < 0.0001, *** *p*-value < 0.001, ** *p*-value < 0.01, and * < 0.05, calculated employing one-way ANOVA with Dunnett’s multiple comparisons tests.

## 3. Results

### 3.1. Single Strain Growths on the Ginseng Extract and Its Components

The strains composing the *in vitro* gut microbiota model demonstrated a remarkable ability to grow on the ginseng extract at a final concentration of 2% *w*/*v*. All of the strains showed an OD_600 nm_ between 3.5 and 4.8, indicating that the extract can sustain their growth (*p*-value < 0.001 vs. mMRS) (Figure 2). Subsequently, the seven strains were also tested for their ability to utilize the single components of the characterized extract. The bacteria exploited the substrates differently, but all of the strains demonstrated their ability to utilize them, reaching an OD_600 nm_ near 1.5 (*p*-value < 0.001 and < 0.01 vs. mMRS) (Figure 2). In particular, galactose is used by the majority of the seven strains; the minimal core strains, except for *B. cellulosilyticus*, are able to grow on galacturonic acid (*p*-value < 0.01); pectin significantly sustains the growth of *L. plantarum* only (*p*-value < 0.05 vs. mMRS) (Figure 2).

### 3.2. Modulation of the In Vitro Reconstructed Human Gut Microbiota in Presence of the Ginseng Extract

We tested the *in vitro* reconstructed human gut microbiota (HGM) setups (previously validated by De Giani et al., 2024 [17]) in the presence of 2% *w*/*v* ginseng as the sole carbon and energy source. The probiotic bacteria reached a maximum OD_600 nm_ of 5.14 ± 0.81 at 32 h, as shown in Figure 3. The minimal core achieved the highest OD_600 nm_ at 48 h of fermentation (4.11 ± 0.19). The growth of the whole community showed the maximum growth at 32 h, reaching an OD_600 nm_ of 3.86 ± 1.02. The modulation of the single strains composing the different configurations showed the prevalence of LP (10^9^ bacterial counts/mL) and BL (10^10^ bacterial counts/mL) among the probiotics (*p*-value < 0.05 vs. time 0) (Figure 4A). Considering the minimal core, BC prevails over the other strains, reaching 10^9^ bacterial counts/mL at 32 h of fermentation (Figure 4B) (*p*-value < 0.05 vs. time 0). In the whole community setup, LP, BL, and BC reached values of bacterial counts/mL in the order of 10^9^, 10^10^, and 10^8^, respectively (Figure 4C). EC is also able to significantly grow when in the presence of all the other strains of the bacterial community, reaching counts of 10^8^ bacteria/mL. However, the growth of the probiotic strains (LP, LA, and BL) is favored overall.

### 3.3. Production of Microbial Metabolites and SCFAs by the In Vitro Reconstructed Human Gut Microbiota

The metabolites derived from the fermentation of the ginseng extract by the probiotics, the minimal core, and the whole bacterial community were extracted and identified. Referring to the probiotic strains (Figure 5A), the first pick (Rt = 9.94 min) was identified as lactic acid. The two picks at Rt = 12.54 and Rt = 13.52 were associated with pentanoic and hexanoic acid, respectively. The other metabolites identified were isocaproic acid (Rt = 16.18) and octanoic acid (Rt = 22.19).

The main metabolites produced by the minimal core in the presence of ginseng extract (Figure 5B) were identified as acetic acid (Rt = 6.96), propionic acid (Rt = 9.94), butyric acid (Rt = 11.59), and pentanoic acid (Rt = 12.54).

Finally, the production of metabolites by the whole bacterial community was analyzed (Figure 5C) showing at the Rt of 9.94 the presence of both lactic and propionic acid (or a mixture of both). The other molecules identified were butyric, pentanoic, hexanoic, and isocaproic acid, associated with an Rt of 11.59, 12.54, 13.52, and 16.18 respectively.

### 3.4. Biological Effects of Bacterial Metabolites and SCFAs on HT-29 Cell Line

The human colorectal cancer HT-29 cell line represents a valuable tool for the evaluation of the beneficial role of probiotics in the human host, as well as for the molecular mechanisms underpinning host–microbe interactions [33].

Intestinal cells showed no difference in their viability after a 48 h treatment with each extract taken into consideration; indeed, at both 0.1 and 0.5 mg/mL, a cellular viability slightly higher than in untreated cells could be observed (Figure 6A).

Given this result, we decided to investigate a possible antioxidant effect, due to the known antioxidant potential derived from ginseng fermentation by probiotics [21]. Cells treated for 4 h with the extract derived from probiotics fermented on ginseng showed a significant reduction in the total reactive oxygen species (ROS) level at 0.1 mg/mL and in cytosolic superoxide anion (O_2_•^−^) content at 0.5 mg/mL.

With regard to the effect of the minimal core bacteria fermented on a 2% ginseng solution, it is possible to observe a statistically significant reduction in the total ROS level at both doses and of the cytosolic superoxide anion at only 0.5 mg/mL. Interestingly, the extracts obtained from the whole community fermented on ginseng caused a 70% reduction in the total ROS level and a 30% reduction in the cytosolic superoxide anion level once given to the cells at the dose of 0.1 mg/mL (Figure 6B,C).

## 4. Discussion

*In vitro* gut microbiota reconstruction models have improved greatly over time, mainly in the rebuilding of digestion stage complexity, the different experimental conditions, and the multitude of ecological parameters to monitor. These models are a useful tool to study the impact of a given diet compound, e.g., prebiotics, or bioactive compound or the addition of probiotics on the human gut microbiota [10,16]. In particular, they can be used to analyze the changes in core microbial groups and selected species together with their metabolites, assaying their diversity and richness in the whole microbial community over time.

In a previous work, we developed a setup of an *in vitro* reconstructed HGM [17], demonstrating the possibility to assay several ecological parameters.

In this paper, we challenged our HGM reconstructed model using a ginseng berry extract, containing bioactive molecules and enriched in polysaccharides, resulting in pectin-based polysaccharides, which we have already demonstrated to possess a prebiotic potential [21].

The *in vitro* HMG setup is based on the findings of a previous study conducted by the same authors of this paper [17]. *E. coli*, *B. cellulosilyticus*, *F. plautii*, and *C. symbiosum* are included in the minimal core due to their classification within the Pseudomonadota, Bacteroidota, and Bacillota phyla, respectively. The minimal core was then implemented by adding the three selected probiotic strains (*L. plantarum*, *L. acidophilus*, and *B. animalis* subsp. *lactis*). The selected configuration reflects the current knowledge about gut microbiota composition *in vivo* [6,34,35]. The majority of the HGM microorganisms belong to the Bacillota and Bacteroidota phyla, while the remaining fall into the Pseudomonadota, Actinomycetota, and Verrucomicrobiota [6].

The novelty of the paper lies in the use of a synthetic human gut *in vitro* microbial community to enhance the understanding of gut bacteria interactions in the presence of a complex extract. The work focuses on analyzing the relationships established within the bacterial community in the presence of complex carbohydrates, such as those found in ginseng extract. The metabolic products derived from the fermentation of the ginseng extract that support the cross-feeding process have been examined.

The ability of the seven synthetic community strains to grow individually on the ginseng extract at 2% concentration was evaluated. The ginseng extract sustained the growth of all of the strains, which showed an OD_600 nm_ between 3.5 and 4.8, with the three probiotic strains showing the highest growth. These data confirm the results of a previous work by De Giani et al. [27]. To better understand which carbohydrate component could promote bacterial growth, we tested the individual polysaccharides found in the extract, including pectin, D-galacturonic acid, and D-galactose. Pectin is considered a next-generation prebiotic due to its beneficial effects, such as regulating glucose and lipid metabolism [36,37] and supporting the growth of beneficial bacteria [38]. The activities of pectin are closely related to its structural characteristics, including monosaccharide composition, molecular weight, degree of esterification (DE), number of glycosidic bonds, ratio of rhamnogalacturonan-I (RG-I) to homogalacturonan (HG), and conformation [39]. Interestingly, of all the strains tested, pectin significantly sustained the growth of *L. plantarum*. This result is in line with previous studies indicating that the survival and growth of *Lactobacillus* sp. is enhanced by pectin [21,40,41]. Notably, D-galacturonic acid significantly sustains the growth of all the minimal core strains except for *B. cellulosilitycus*. Most of the seven strains showed the ability to use D-galactose as a carbon source.

After evaluating individual strains on the ginseng extract, we tested the *in vitro* reconstructed HGM setups [17] using 2% *w*/*v* ginseng as the only carbon source. The exponential growth phase of the three configurations (probiotics, minimal core, and whole community) started at 8 h of incubation and reached the stationary phase at around 32 h. Probiotic bacteria reached the highest growth, while the minimal core and the whole community showed almost the same growth. The modulation of the single strains showed the prevalence of *L. plantarum* and *B. animalis* subsp. *lactis* among the probiotics, while *B. cellulosilyticus* prevailed over the other strains in the minimal core setup. In the presence of both probiotics and minimal core strains, *L. plantarum*, *B. animalis* subsp. *lactis*, and *B. cellulosilyticus* reached the highest values of bacterial counts/mL. Interestingly, *E. coli* was able to grow significantly when in the presence of all the other strains of the bacterial community.

The *Bacteroides* genus is known as a primary degrader in the metabolism of complex carbohydrates producing several metabolites such as acetate and propionate [2,41].

In contrast, the *Bifidobacterium* genus more commonly metabolizes smaller oligosaccharides, producing lactate as the main metabolite [2,42]. As expected, due to the presence of both *B. animalis* subsp. *lactis* and *L. plantarum*, lactic acid was detected in the presence of probiotics, both alone and in combination with the minimal core strains. Indeed, *L. plantarum* is an heterofermenter [43]. Acetate was detected in the minimal core setup, while propionate is present in the minimal core and the whole community. Acetate and lactate can then be metabolized to produce butyrate by butyrate-producing bacteria such as Clostridia of the cluster IV and XIV species like *Clostridium symbiosum* and *Flavonifractor plautii* [2]. Indeed, in our work, butyrate has been identified in the presence of the minimal core strains alone and with the probiotics. *L. plantarum* and *E. coli* are also able to degrade the polysaccharides present in the extract and the oligosaccharides produced during fermentation.

All of the metabolites produced by the bacteria have an important effect on the host, in addition to their role in the metabolic interplay of the HGM. Antitumoral, anti-inflammatory, and antioxidant effects are some of the properties of the beneficial metabolites produced by the intestinal bacterial community [6]. The metabolites obtained from ginseng extract fermentation by probiotics, the minimal core, or the whole community do not affect intestinal cell viability. Since ginseng extract is well known for its antioxidant properties [20,21,22], we focused our attention on the antioxidant effect. Our results highlighted a significant reduction in total ROS when intestinal cells were exposed to 0.1 mg/mL of metabolites derived from ginseng extract fermentation by probiotics and the minimal core. Moreover, these two experimental groups caused a slight reduction in superoxide anion content when given to the cells at a final concentration of 0.5 mg/mL. Remarkably, 0.1 mg/mL of bacterial metabolites produced from the whole community in the presence of ginseng determined a major reduction in both total ROS and cytosolic superoxide anion levels. Indeed, short-chain fatty acids (SCFAs), in particular butyrate and acetate, are able to modulate oxidative stress [44]. Human colonocytes exposed to butyrate showed a notable decrease in DNA damage caused by H_2_O_2_ [45]. In addition, butyrate increased the activity of glutathione S-transferase in human colon carcinoma HT-29 cells [46].

This research represents an example of creating a rationally designed microbiota-based model for understanding gut microbial interactions in the presence of different prebiotic compounds to complement missing or underrepresented functions that, in *in vivo* situations, cannot be deeply studied. However, this kind of study is characterized by several limitations. In fact, there are several aspects that should be reinforced such as the stability of the whole microbial community during the time of the experiment or the technology used to include host cells. Another limitation is the use of a few strains, showing a reductionistic approach with respect to *in vivo* conditions where there is a more complex ecosystem.

On the other hand, this approach permits us to investigate the influence on gut microbiota of a vast variety of factors such as compounds introduced by the diet, microbial pathogens, bioactive compounds, and pharmaceutical substances and to better mimic this ecosystem for the development of strategies aimed to reduce the imbalance of the gut microbial community with probiotic supplementation.

## 5. Conclusions

In conclusion, this study shows that the fermentation of ginseng extract with probiotic bacteria in the presence of a reconstructed minimal HGM mediates an antioxidant effect on the HT-29 human intestinal cell line. Therefore, these results highlight a synergism between the host and its own intestinal microbiota and the significant role of administering a probiotic treatment alongside a suitable prebiotic to enhance beneficial antioxidant effects.

## Figures and Tables

**Figure 1 microorganisms-13-00192-f001:**
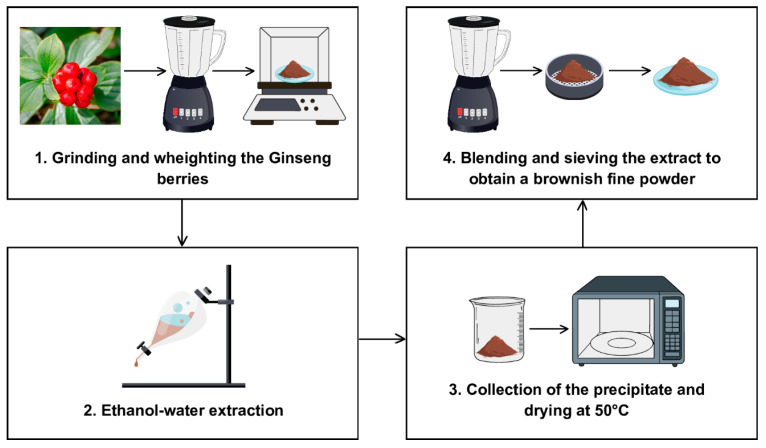
The extraction process for obtaining the extract from the berries of *Panax ginseng*.

**Figure 2 microorganisms-13-00192-f002:**
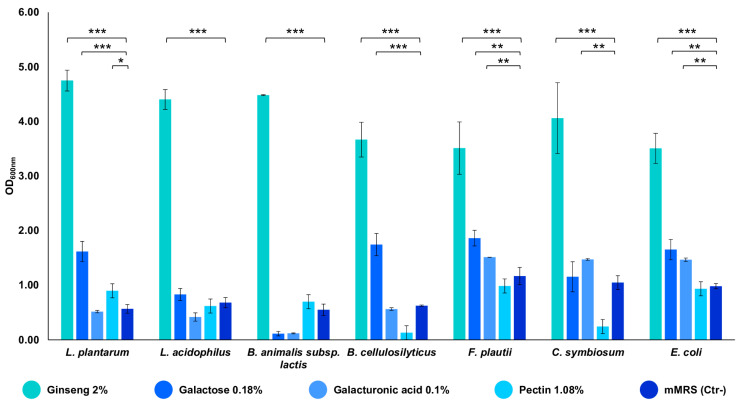
Growths of the single bacterial strains on ginseng extract and its components. The growth is presented as the mean value of OD_600 nm_ ± SE in the presence of 2% ginseng extract, 0.18% galactose, 0.1% galacturonic acid, 1.08% pectin, and mMRS (Ctr- medium). The statistical significance was calculated via Student’s *t* test: * *p*-value < 0.05, ** *p* < 0.01, *** *p* < 0.001.

**Figure 3 microorganisms-13-00192-f003:**
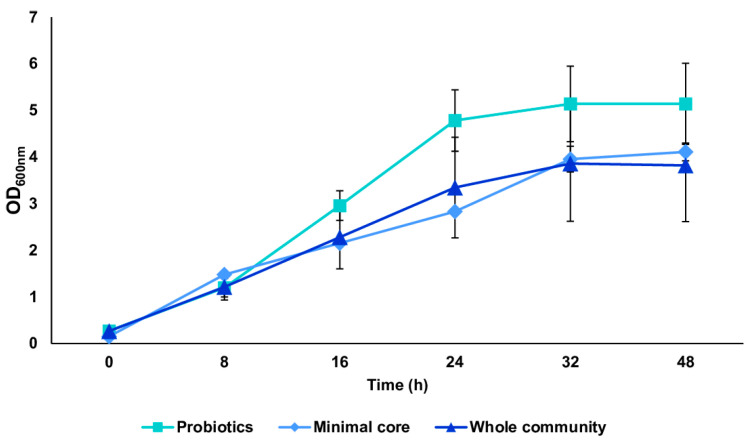
Growth curves of probiotics, minimal core, and the whole community. Growth curves (mean value of OD_600 nm_ ± SE during time) of the three conditions: probiotics, minimal core, and the whole community, in the presence of 2% ginseng extract in batch fermentation.

**Figure 4 microorganisms-13-00192-f004:**
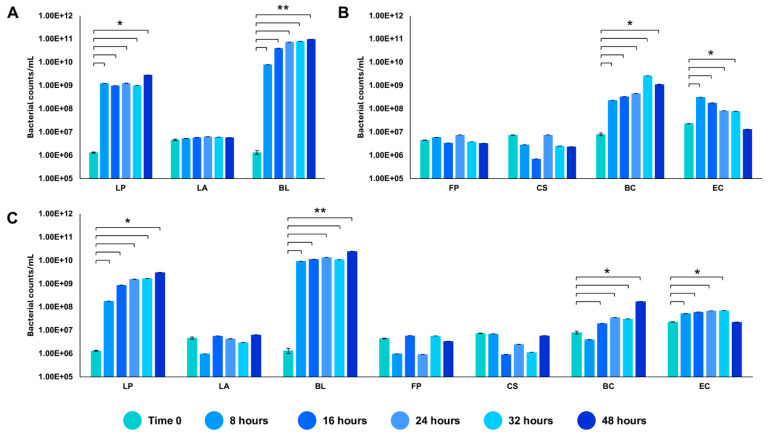
Modulation of the single strains through qRT-PCR. Values of bacterial counts/mL reached by the probiotics (**A**), the minimal core strains (**B**), and the whole community (**C**) during growth on 2% ginseng extract. The statistical significance was calculated using Student’s *t* test: * *p*-value < 0.05, ** *p* < 0.01.

**Figure 5 microorganisms-13-00192-f005:**
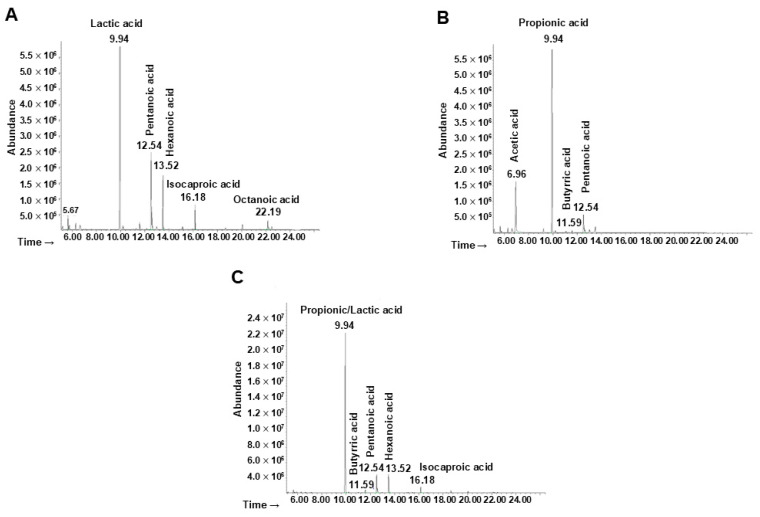
Analyses of bacterial metabolites by GC-MSD after ginseng extract fermentation. Chromatogram of metabolites produced by (**A**) probiotics, (**B**) minimal core strains, and (**C**) the whole community.

**Figure 6 microorganisms-13-00192-f006:**
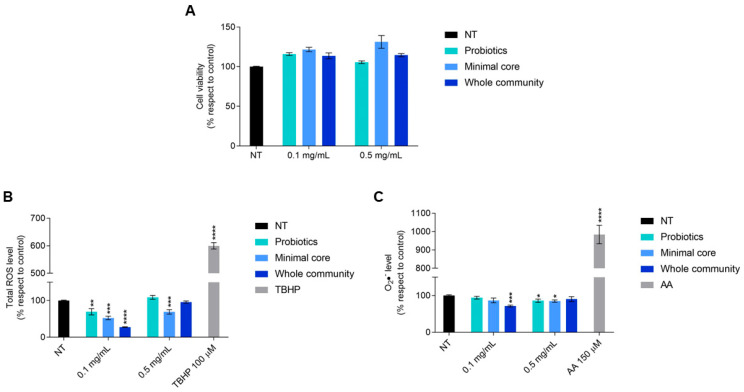
Viability of HT-29 cells exposed for 48 h to the extracts derived from probiotics, minimal core bacteria, and the whole community fermented on a 2% ginseng extract. (**A**) Oxidative stress response of HT-29 after a 4 h exposure to metabolites deriving from probiotics, minimal core bacteria, and the whole community fermented on ginseng extracts. (**B**) Panel shows the total ROS level measured by H_2_DCFDA. (**C**) Panel shows the cytosolic superoxide anions (O_2_•^−^) content measured by DHE. TBHP 100 µM and antimycin A (AA) 150 µM were used as a positive control for H_2_DCFDA and DHE, respectively. Data are represented as the mean ± standard error (SE). Statistically significant results are highlighted with **** *p*-value < 0.0001, *** *p*-value < 0.001, ** *p*-value < 0.01, and * *p* < 0.05.

**Table 1 microorganisms-13-00192-t001:** Characterization of ginseng extract.

Component	*P. Ginseng* Extract (%)
Pectin-based polysaccharides Galactose Galacturonic acid	54.30 9.01 4.84
Ginsenosides Ginsenosides Re	10.00 2.00
Proteins	0.98
Polyphenols	0.72
Unidentified molecules	34

**Table 2 microorganisms-13-00192-t002:** Bacterial strains used in the study.

Strain	Source	Abbreviation
*Lactobacillus acidophilus* PBS066 (DSM 24936)	Human	LA
*Lactiplantibacillus plantarum* PBS067 (DSM 24937)	Human	LP
*Bifidobacterium animalis* subsp. *lactis* BL050 (DSM 25566)	Human	BL
*Bacteroides cellulosilyticus* CL02T12C19, HM-726	Human	BC
*Clostridium symbiosum* WAL-14673, HM-319	Human	CS
*Flavonifractor plautii* (previously *Clostridium orbiscindens*) 1_3_50AFAA, HM-303	Human	FP
*Escherichia coli* ATCC 25922	Human	EC

## Data Availability

The original contributions presented in this study are included in the article/Appendix A. Further inquiries can be directed to the corresponding author.

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
