# Peer review of "Beneficial Effects of In Vitro Reconstructed Human Gut Microbiota by Ginseng Extract Fermentation on Intestinal Cell Lines"

_microorganisms, 2025, doi:10.3390/microorganisms13010192_

Round 1
Reviewer 1 Report
Comments and Suggestions for Authors
This work is particularly interesting, well-written, and supported by a clearly outlined methodology that enhances data reproducibility. The figures are valuable and highly legible.
- I believe the discussion should explicitly and thoroughly address the inherent limitations and potential biases arising from constructing a microbiota in vitro rather than in vivo for the examination of outcomes of interest. It would also be important to clarify what additional steps are necessary for validating these initial results in future research efforts.
Minor considerations:
- In the introduction, I suggest including a reference to the emerging role of fungi as beneficial elements for gut microbiota. For example, Hericium erinaceus could be mentioned. A recent review on the role of H. erinaceus in gastrointestinal diseases and a recent ex vivo study demonstrating initial anti-inflammatory effects in IBD models might be relevant here.
- Terms such as in vitro (e.g., in the title) should be italicised.
Author Response
REVIEWER 1
This work is particularly interesting, well-written, and supported by a clearly outlined methodology that enhances data reproducibility. The figures are valuable and highly legible.
I believe the discussion should explicitly and thoroughly address the inherent limitations and potential biases arising from constructing a microbiota in vitro rather than in vivo for the examination of outcomes of interest. It would also be important to clarify what additional steps are necessary for validating these initial results in future research efforts.
Reply: The Discussion has been improved with the points suggested by the Reviewer.
Minor considerations:
In the introduction, I suggest including a reference to the emerging role of fungi as beneficial elements for gut microbiota. For example, Hericium erinaceus could be mentioned. A recent review on the role of H. erinaceus in gastrointestinal diseases and a recent ex vivo study demonstrating initial anti-inflammatory effects in IBD models might be relevant here.
Reply: During the writing of the introduction of the manuscript we had taken into consideration to include also emerging role of fungi as beneficial elements for gut microbiota, however we decided to refer to bacteria because our data provided results only on this aspect and so could be discussed with literature data. We appreciate for the suggestion, but we retain that this inclusion can generate a sort of confusion to the reader and we prefer not include this aspect.
Terms such as in vitro (e.g., in the title) should be italicised.
Reply: Done, and it has been corrected in all the text.
Reviewer 2 Report
Comments and Suggestions for Authors
REVIEW
Dear authors,
The work presents evidence of the use of Ginseng extract fermentation on an in vitro system simulating the human intestinal microbiota, which is supported by a previous study that supports the work on probiotic microorganisms and core microorganisms used in the present work. The focus on the production of short chain fatty acids (SCFA) and the reduction of the production of reactive oxygen species (antioxidant effect), demonstrate the potential benefits of Ginseng in the modulation of a microorganism system on an intestinal cell line at the in vitro level.
Please consider the following comments to improve the content of your manuscript before publication.
1. Line 2: write the term in italics “in vitro”.
2. Lines 14, 15, 17, 20-23: write the scientific name of microorganisms in cursive “L. plantarum, B. animalis subsp. lactis, B. cellulosilyticus”, as well as “P. ginseng” and the term “in vitro”.
3. Line 63: the abbreviation “HGM” has already been used previously on line 37, there is no need to indicate it again.
4. Line 157: write in subscript “OD600nm”.
5. Lines 202-203: write the term in italics “in vitro”
6. Line 214: write in subscript “H2DCFDA”.
7. Line 219: write the unit correctly “mL”.
8. Lines 229-231: replace the Bradford technique reference with numbering.
9. Line 245: write the term in italics “in vitro”.
10. Line 309: write the unit correctly “mL”.
11. Figures 6A, 6B y 6C write the unit correctly “mL”.
12. Lines 332, 339, 345: write the term in italics “in vitro”.
13. Lines 420-425: add the corresponding text to the section 5. Conclusions.
14. Could Ginseng extract stimulate the production of plantaricins (L. plantarum) or antimicrobial peptides from probiotic strains?
15. It is necessary to demonstrate the cytokine production profile of HT-29 cells when they are in the presence of Ginseng extract, as well as the different combinations of microorganisms to determine if an immunomodulatory effect occurs, especially with microorganisms that are more immunogenic such as Escherichia coli ATCC 25922.
16. The results are promising, however, they must be transferred to the in vivo model to evaluate the safety of the Ginseng extract and demonstrate its antioxidant effect. What model would you propose?
Please amend the requested comments and submit the revision file.

Author Response
REVIEWER 2
The work presents evidence of the use of Ginseng extract fermentation on an in vitro system simulating the human intestinal microbiota, which is supported by a previous study that supports the work on probiotic microorganisms and core microorganisms used in the present work. The focus on the production of short chain fatty acids (SCFA) and the reduction of the production of reactive oxygen species (antioxidant effect), demonstrate the potential benefits of Ginseng in the modulation of a microorganism system on an intestinal cell line at the in vitro level.
Please consider the following comments to improve the content of your manuscript before publication.
- Line 2: write the term in italics “in vitro”.
Reply: Done.
- Lines 14, 15, 17, 20-23: write the scientific name of microorganisms in cursive “L. plantarum, B. animalissubsp. lactis, B. cellulosilyticus”, as well as “P. ginseng” and the term “in vitro”.
Reply: Done.
- Line 63: the abbreviation “HGM” has already been used previously on line 37, there is no need to indicate it again.
Reply: Done.
- Line 157: write in subscript “OD600nm”.
Reply: Done.
- Lines 202-203: write the term in italics “in vitro”
Reply: Done.
- Line 214: write in subscript “H2DCFDA”.
Reply: Done.
- Line 219: write the unit correctly “mL”.
Reply: Done.
- Lines 229-231: replace the Bradford technique reference with numbering.
Reply: Done.
- Line 245: write the term in italics “in vitro”.
Reply: Done.
- Line 309: write the unit correctly “mL”.
Reply: Done.
- Figures 6A, 6B y 6C write the unit correctly “mL”.
Reply: Done.
- Lines 332, 339, 345: write the term in italics “in vitro”.
Reply: Done.
- Lines 420-425: add the corresponding text to the section5. Conclusions.
Reply: The corresponding text to the Section Conclusion has been added.
- Could Ginseng extract stimulate the production of plantaricins (L. plantarum) or antimicrobial peptides from probiotic strains?
Reply: We have not investigated the production of antimicrobial peptides from Ginseng extract in particular, because the focus of the work was mainly the production of metabolites produced from polysaccharides contained in the Ginseng extract supplied as prebiotic source.
- It is necessary to demonstrate the cytokine production profile of HT-29 cells when they are in the presence of Ginseng extract, as well as the different combinations of microorganisms to determine if an immunomodulatory effect occurs, especially with microorganisms that are more immunogenic such as Escherichia coliATCC 25922.
Reply: We have not investigated the production of cytokine of HT-29 cells in this paper, because we have evaluated the antioxidant effect. These data are available from literature. In fact, several papers had reported that P. ginseng extract exhibit anti-inflammatory and immunomodulatory effects in vitro and in vivo.
For example, Lee and coworkers (Lee, I. A., Hyam, S. R., Jang, S. E., Han, M. J., & Kim, D. H. 2012, Ginsenoside Re ameliorates inflammation by inhibiting the binding of lipopolysaccharide to TLR4 on macrophages. Journal of agricultural and food chemistry, 60(38), 9595–9602), showed that in peritoneal macrophages ginsenoside Re suppress the expression of proinflammatory cytokines, by inhibiting LPS binding to TLR4; while it reduces the expression of proinflammatory cytokines, such as TNF-α and IL-1β, and enhances the expression of the anti-inflammatory cytokine IL-10 in TNBS-induced colitis in mice. Moreover, Ahn and colleagues (Ahn, S., Siddiqi, M. H., Aceituno, V. C., Simu, S. Y., & Yang, D. C. 2016. Suppression of MAPKs/NF-κB Activation Induces Intestinal Anti-Inflammatory Action of Ginsenoside Rf in HT-29 and RAW264.7 Cells, Immunological Investigations, 45(5), 439–449), had demonstrated a beneficial immunomodulatory effect of ginsenoside Rf on TNF-α-stimulated intestinal epithelial cells (HT-29) and mouse macrophage cells (RAW264.7).
Finally, the purified polysaccharides from P. ginseng showed intestinal anti-inflammatory effects in DSS-induced rats by regulating gut microbiota and mTOR dependent autophagy (Wang, D., Shao, S., Zhang, Y., Zhao, D., & Wang, M. (2021). Polysaccharides from Panax ginseng C. A. Meyer in Improving Intestinal Inflammation: Modulating Intestinal Microbiota and Autophagy. Frontiers in immunology, 12). These authors also demonstrated that the purified polysaccharides could regulate the structure of gut microbiota and reduce the abundance of Gram-negative bacteria that produce LPS. In fact, ginseng polysaccharides play a pivotal role in influencing intestinal metabolism, modulating gut microbiota, as well as prominently reinforcing the growth of Lactobacillus and Bacteroides, two crucial bacterial species that are able to metabolize these compounds.
- The results are promising, however, they must be transferred to the in vivomodel to evaluate the safety of the Ginseng extract and demonstrate its antioxidant effect. What model would you propose?
Reply: These data are fundamentals just to be transferred to an in vivo model. We can propose both an in vivo study on recruited human subjects supplying the Ginseng extract (because this extract is commercialized) or an animal model (mouse) supplying the Ginseng extract and evaluating the antioxidant effect on cells from the animal.